# A Nomogram Built on Clinical Factors and CT Attenuation Scores for Predicting Treatment Response of Acute Myeloid Leukemia Patients

**DOI:** 10.3390/biomedicines13010198

**Published:** 2025-01-15

**Authors:** Linna Liu, Wenzheng Lu, Li Xiong, Han Qi, Robert Peter Gale, Bin Yin

**Affiliations:** 1Clinical Research Center, Jiangnan University Medical Center, 68 Zhongshan Road, Wuxi 214002, China; liulinna0728@163.com (L.L.); xl1797200376@163.com (L.X.); qihan012023@163.com (H.Q.); 2Department of Radiology, Affiliated Hospital, Jiangnan University, No.1000, Hefeng Road, Wuxi 214000, China; luwz2022@163.com; 3Haematology Research Centre, Department of Immunology and Inflammation, Imperial College London, London SW7 2AZ, UK; robertpetergale@gmail.com

**Keywords:** acute myeloid leukemia, computed tomography, treatment response, nomogram

## Abstract

**Background:** Acute myeloid leukemia (AML) is an aggressive cancer with variable treatment responses. While clinical factors such as age and genetic mutations contribute to prognosis, recent studies suggest that CT attenuation scores may also predict treatment outcomes. This study aims to develop a nomogram combining clinical and CT-based factors to predict treatment response and guide personalized therapy for AML patients. **Methods**: This retrospective study included 74 newly diagnosed AML patients who underwent unenhanced abdominal CT scans within one week before receiving their first induction chemotherapy. Clinical biomarkers of tumor burden were also collected. Patients were classified into two groups based on treatment response: complete remission (CR; *n* = 24) and non-complete remission (NCR; *n* = 50). Multivariable logistic regression was used to identify independent predictors of treatment response. Predictive performance was evaluated using receiver operating characteristic (ROC) curves, and model consistency was assessed through calibration and decision curve analysis (DCA). **Results**: Significant differences in hemoglobin (Hb), platelets (Plt), and CT attenuation scores were observed between the CR and NCR groups (all *p* < 0.05). Multivariable logistic regression identified Hb, Plt, and CT attenuation scores as independent predictors of treatment response. A nomogram incorporating these factors demonstrated excellent predictive performance, with an area under the curve (AUC) of 0.912 (95% CI: 0.842–0.983), accuracy of 0.865 (95% CI: 0.765–0.933), sensitivity of 0.880 (95% CI: 0.790–0.970), and specificity of 0.833 (95% CI: 0.684–0.982). The CR nomogram displayed significant clinical value and excellent goodness of fit. **Conclusions**: The nomogram, which incorporates Hb, Plt, and CT attenuation scores, provides valuable insights into predicting treatment response in AML patients. This model offers strong discriminatory ability and could enhance personalized treatment planning and prognosis prediction for AML.

## 1. Introduction

Acute myeloid leukemia (AML) is a highly heterogeneous hematologic malignancy characterized by the abnormal proliferation of leukemic cells in the bone marrow and peripheral blood, which suppresses normal hematopoiesis, leading to severe complications such as anemia, infection, and hemorrhage [1]. Currently, AML treatment typically follows the “7+3” regimen, combining *Cytarabine* (Ara-C) inhibiting DNA synthesis and *Daunorubicin*, inducing DNA damage. *Etoposide*, a topoisomerase II inhibitor, can enhance this regimen’s effect. For relapsed or refractory cases, *Gemtuzumabozogamicin*, a toxin-conjugated monoclonal antibody targeting the *CD33* antigen, is often used in combination with chemotherapy. These therapies provide diverse treatment options for both induction therapy and relapsed AML [2,3,4,5,6]. However, the evaluation of treatment response often relies on invasive procedures such as bone marrow aspiration [7], which cause significant discomfort to patients and are associated with risks of infection and hemorrhage [8]. Therefore, the development of non-invasive techniques that can replace these traditional methods has become an urgent need in clinical practice.

In recent years, the application of imaging technologies in the management of AML has gradually gained attention [9,10]. Traditionally, imaging techniques such as PET, MRI, and CT have been primarily used for the diagnosis and treatment evaluation of solid tumors [11,12,13]. By reflecting changes in tissue structure, metabolism, and affected regions, these technologies have become essential tools in cancer diagnosis and treatment. Among them, PET/CT may be valuable as a non-invasive tool for early assessment of treatment response and may provide prognostic value for survival in patients with AML [14]. MRI provides high-resolution soft tissue imaging, showing efficacy in detecting bone marrow involvement and leukemic infiltration [15,16]. CT, as a quantitative imaging technique, offers the advantages of high resolution and rapid scanning, making it widely used in tumor diagnosis, treatment assessment, and follow-up, particularly in the management of solid tumors. Some studies have focused on the use of CT to differentiate AML patients from the general population [17]. However, research on the role of CT in evaluating treatment response in AML is still limited.

Therefore, this study aims to investigate whether CT attenuation value can be used to predict the efficacy response in AML patients.

## 2. Material and Methods

### 2.1. Ethics Statement

The study was approved by the Ethics Committee of Wuxi Second People’s Hospital, Jiangnan University Medical Center (No. Y-20), and conducted in accordance with the tenets of the Declaration of Helsinki. Due to the retrospective nature of the study, informed consent was waived for all patients.

### 2.2. Patients

A total of 112 patients with AML diagnosed from April 2019 to June 2023 underwent abdominal CT scans within one week prior to the treatment. The inclusion criteria for the study were as follows: patients who met the pathological diagnostic criteria for AML completed initial treatment and had complete clinical data and CT imaging records. Exclusion criteria were patients with incomplete treatment courses, incomplete clinical or imaging data, or other severe comorbidities. A total of 74 patients were finally included in the study, and their CT scan data and clinical measurements were retrospectively analyzed (Figure 1).

### 2.3. Clinical Data

The basic clinical characteristics of the patients were collected, including age, sex, and hematological parameters before and after treatment. These hematological parameters included white blood cells (WBCs), hemoglobin (Hb), platelets (Plt), neutrophils (NEs), and bone marrow blasts (BMBs). The clinical treatment details and hematological parameters of AML patients before treatment were collected, and the data closest to the pretreatment CT scans were selected for analysis. The data were obtained from the ‘patient’s medical records and reviewed by experts. The treatment regimens administered to the patients included the standard “7+3” induction chemotherapy containing Cytarabine (Ara-C) and Daunorubicin. High-risk patients also received additional treatments such as Etoposide and Gemtuzumabozogamicin in accordance with the guidelines for the treatment of relapsed or refractory AML cases. Consolidation therapy was conducted with a high dose of Ara-C or hematopoietic stem cell transplantation for the eligible patients [5,6]. Complete remission (CR) was defined as the normalization of bone marrow and peripheral blood with ≤5% bone marrow blasts, granulocyte count greater than 1 × 10^3^/μL, platelet count greater than 100 × 10^3^/μL, and normal differential. Patients who did not achieve CR were divided into non-CR (NCR) groups [18].

### 2.4. CT Scanning

All patients underwent axial abdominal CT scans prior to completion of treatment. The scans were performed on multi-slice CT scanners, including the Lightspeed VCT (GE Medical Systems, Waukesha, WI, USA) and SOMATOM Definition Flash (Siemens Healthcare, Forchheim, Germany). Scanning parameters were as follows: slice thickness 5 mm, slice interval 1 mm, tube voltage 120 kV, and tube current 200 mA.

### 2.5. Image Evaluation

CT attenuation scores (Hounsfield units, HU) were measured in the bone window by delineating a region of interest (ROI) larger than 100 mm^2^ on both sides of the iliac bone at the level of the posterior superior iliac spine [17]. The ROI was delineated to avoid cortical bone, sclerosis, bone islands, and fracture areas. In addition, the craniocaudal length and thickness of the spleen were measured to assess the presence of splenomegaly [19]. Two radiologists (Reader 1 and Reader 2), with 5 and 10 years of experience in abdominal imaging, respectively, independently measured the CT parameters of lesions without knowing the treatment outcomes. All measurements were performed independently, and the mean values of the measurements were used for data analysis.

### 2.6. Statistical Analysis

Data were processed using statistical software such as SPSS (version 26.0) or R (version 4.0). Normality tests were performed on the data. For normally distributed continuous variables, results are presented as mean ± standard deviation (x ± s) and were compared using the *t*-test. Non-normally distributed continuous variables are expressed as median (first quartile, third quartile) [M (Q1, Q3)] and were compared using the Mann–Whitney U test. Categorical variables are presented as counts (percentages) [*n* (%)] and were compared using the χ^2^ test. The intra-class correlation coefficient (ICC) was used to evaluate the inter-observer agreement of CT parameter measurements between the two radiologists. Univariate and multivariate logistic regression analyses were performed to determine the independent predictors. Factors with a *p*-value < 0.05 were included in the multivariate analysis. A nomogram was then constructed based on the independent predictors identified through multivariate logistic regression analysis. The performance of the model was assessed using receiver operating characteristic (ROC) curves to calculate the area under the curve (AUC), accuracy, sensitivity, and specificity. The optimal cut-off value was determined by the maximum Youden index. The performance of the models was compared using the DeLong test. A calibration curve was performed to assess the consistency (goodness of fit) of the model between actual and predicted treatment response risks. Decision curve analysis (DCA) was performed to quantify clinical utility. All statistical tests were two-tailed and a *p*-value of less than 0.05 was considered statistically significant.

## 3. Results

A total of 74 patients were included in this study and their clinical characteristics are detailed in Table 1. The median age of the patients was 70.50 years (interquartile range [IQR], 54.25–80.00). There were no significant differences in age or gender between the CR and NCR groups (*p* > 0.05). Hematological parameters varied between the two groups, with Hb and Plt levels significantly higher in the CR group compared to the NCR group (*p* < 0.05); however, there were no significant differences in WBC, NE, or BMB levels (*p* > 0.05). Regarding imaging parameters, CT values were significantly higher in the CR group than in the NCR group (*p* < 0.05), whereas no significant differences were observed in spleen length and thickness (*p* > 0.05). Figure 2 shows the two sets of representative images obtained from CT for the CR and NCR groups, respectively.

Univariate and multivariable logistic regression analyses identified Hb, Plt, and CT values as independent predictors of treatment response (Table 2). A clinical model was constructed based on Hb and Plt, while an imaging model was constructed based on CT values. In addition, a combined model including all three predictors (Hb, Plt, and CT values) was developed and visualized as a nomogram (Figure 3). The performance of the three models was evaluated and showed that the AUC of the clinical model was 0.762 (95% CI: 0.639–0.885), the AUC of the CT model was 0.882 (95% CI: 0.804–0.960) and the AUC of the nomogram was 0.912 (95% CI: 0.842–0.983). These results indicate that the nomogram model had the best predictive performance, also demonstrating high accuracy (0.865, 95% CI: 0.765–0.933), sensitivity (0.880, 95% CI: 0.790–0.970), and specificity (0.833, 95% CI: 0.684–0.982) (Table 3 and Figure 4). The DeLong test showed that the diagnostic performance of the nomogram was significantly better than that of the clinical model (*p* = 0.005), but there was no significant difference compared to the imaging model (*p* = 0.318). In addition, calibration curves and decision curve analysis were used to assess the performance of the nomogram (Figure 5), and the results indicated that the nomogram had good clinical utility and potential for predicting treatment response (Figure 6).

## 4. Discussion

In this study, we investigated the significance of the attenuation value of the unenhanced CT in assessing treatment response in AML patients. We found that CT value was an independent risk factor for predicting treatment response and had good predictive efficiency. The nomogram established by combining clinical indicators can further improve the predictive efficiency, which has good clinical practical value.

Although bone marrow aspiration is the standard for assessing response and residual disease, it is an invasive assessment method due to the risk of infection [7,8]. Non-invasive imaging techniques are currently used to assess treatment response and outcome [14,16,20,21]. However, there are few reports on the CT manifestations of bone marrow infiltration in AML patients, and even more so, there is no relevant report on the study of the CT value in predicting treatment response in AML patients. Therefore, we investigated the value of non-contrast CT in assessing treatment response in AML. According to our study, there is a significant difference in CT attenuation value between AML patients in the CR group and the NCR group, and the CT attenuation value of bone marrow in the CR group will increase compared with the NCR group. Logistic regression showed that CT value was an independent predictor of treatment response with good diagnostic performance (AUC: 0.882).

We also evaluated the value of clinical indicators in assessing treatment response. Our results showed that Hb and Plt were significantly higher in the CR group than in the NCR group. The clinical model (Hb and Plt) constructed based on multivariable regression analysis showed good diagnostic performance in predicting treatment response in AML, with an AUC of 0.762. These results support the efficacy of hematological indicators as prognostic factors for AML, suggesting that higher Hb and Plt levels may be associated with better treatment response [22,23]. Our study showed that the nomogram combining significant clinical hematological indicators (Hb and Plt) with quantitative imaging values (non-enhanced CT values) significantly improved the predictive performance for treatment response in AML, with an AUC of 0.912. This suggests that the integration of clinical and imaging indicators may further improve management strategies for AML patients.

Despite the promise of this study, there are some limitations. Firstly, this is a retrospective, single-center study with a limited sample size, which might not entirely exclude the possible bias in patient selection, although the data of our cohort of patients were collected consecutively during a fixed period of time. Secondly, incorporating AML-associated genomic data would refine our results; however, it would be complicated to quantify the results of genetic analyses. Future research is warranted to focus on multi-center, large-sample prospective studies and explore the potential of combining imaging data with artificial intelligence technologies to further improve predictability.

## 5. Conclusions

This study demonstrates that integrating imaging and clinical data into a nomogram can achieve a better non-invasive prediction of treatment response in AML.

## Figures and Tables

**Figure 1 biomedicines-13-00198-f001:**
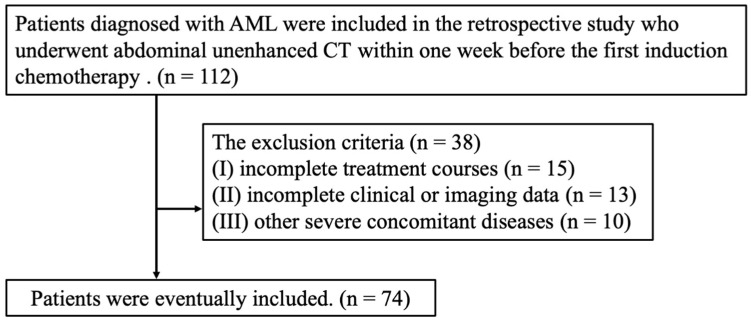
Flow chart of inclusion and exclusion of patients.

**Figure 2 biomedicines-13-00198-f002:**
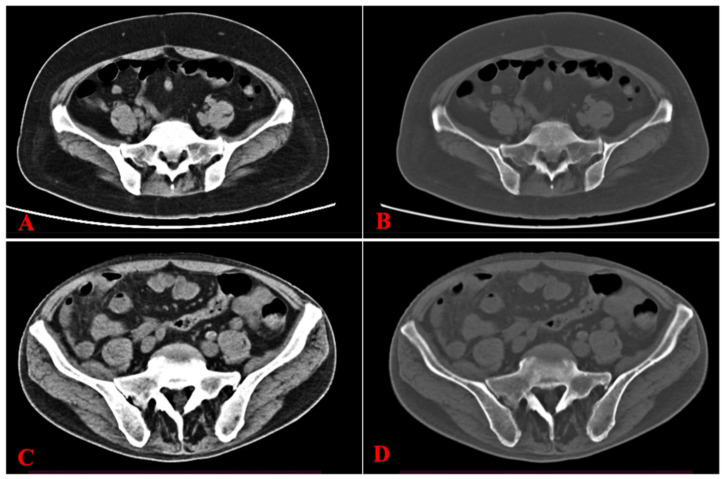
Pretreatment CT images of representative AML patients who achieved complete remission (CR) and those who did not achieve complete remission (NCR). (**A**): Axial unenhanced CT with an abdominal window, and (**B**): a bone window showed an increased bone marrow density of both iliac bones (CT: 212.03 HU). (**C**,**D**): A representative AML patient who achieved NCR after the therapy. (**C**): Axial unenhanced CT with an abdominal window, and (**D**): a bone window showed a lower bone marrow density of both iliac bones (CT: 116.08 HU). AML, acute myeloid leukemia; CR, complete remission; NCR, non-CR.

**Figure 3 biomedicines-13-00198-f003:**
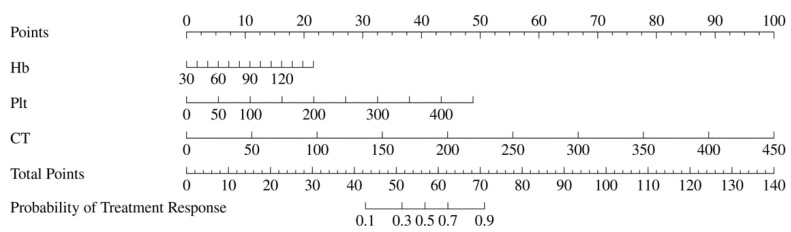
The nomogram that was employed to forecast the probability of the complete remission for patients with AML. The points by which two factors (Hb, Plt, and CT), respectively, make a vertical line to the topmost line are added to obtain total points; then, the total points correspond to the probability of treatment response of the AML patient with treatment response of the bottom line. AML, acute myeloid leukemia; Hb, hemoglobin; Plt, platelets; CT: CT attenuation scores.

**Figure 4 biomedicines-13-00198-f004:**
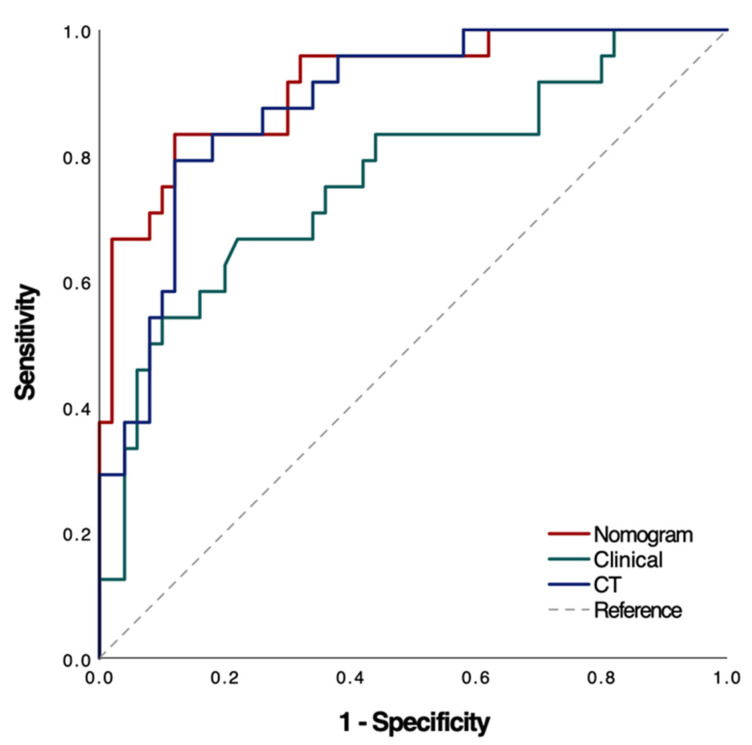
ROC curves of the models in predicting the complete remission. The nomogram was composed of clinical test indicators (Hb and Plt) and CT attenuation scores. Clinical was composed of Hb and Plt in clinical laboratory indicators. The CT model was composed of the measured CT attenuation scores.

**Figure 5 biomedicines-13-00198-f005:**
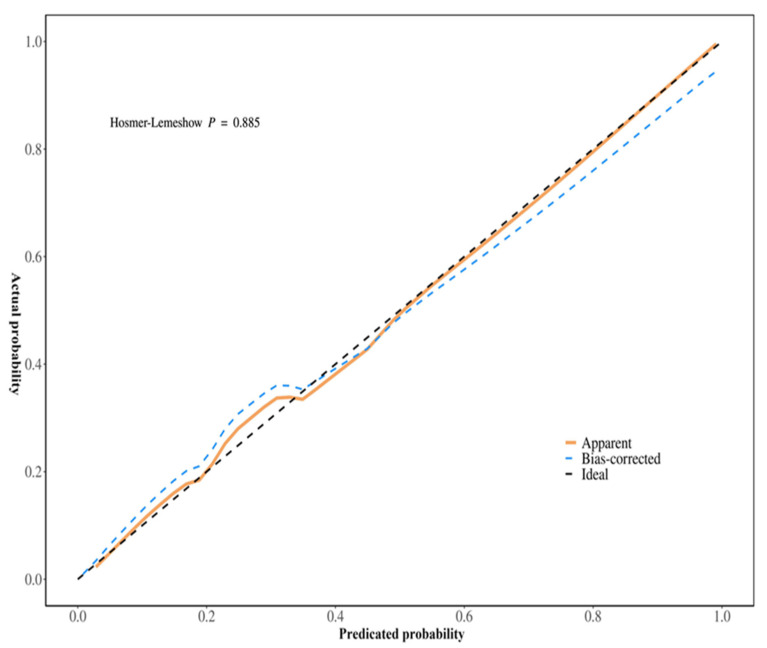
The calibration curve of the nomogram. The Hosmer–Lemeshow goodness-of-fit test is a method used to assess the goodness of fit of binary logistic regression models. *p* = 0.885 represented good fitting efficacy.

**Figure 6 biomedicines-13-00198-f006:**
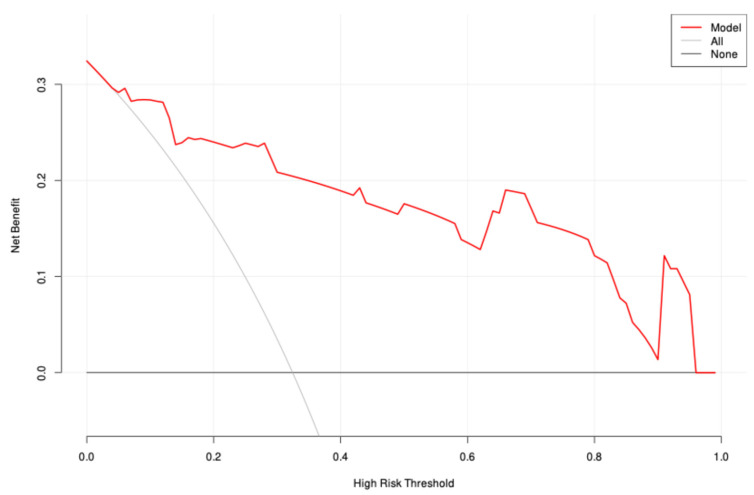
Decision curve analyses for the simple-to-use model predicting complete remission in the nomogram.

**Table 1 biomedicines-13-00198-t001:** Clinical characteristics of the patients.

Variables	Total (*n* = 74)	CR (*n* = 24)	NCR (*n* = 50)	*p*
Age, M (Q_1_, Q_3_)	70.50 (54.25, 80.00)	68.50 (58.75, 78.50)	72.00 (54.00, 81.00)	0.959
Gender, *n* (%)				0.135
Male	28 (37.84)	12 (50.00)	16 (32.00)	
Female	46 (62.16)	12 (50.00)	34 (68.00)	
WBC (×10^9^/L), M (Q_1_, Q_3_)	10.36 (2.51, 44.54)	32.74 (4.25, 69.65)	7.35 (2.50, 32.75)	0.120
Hb (g/L), Mean ± SD	87.77 ± 24.74	99.00 ± 24.94	82.38 ± 23.00	0.006 *
Plt (×10^9^/L), M (Q_1_, Q_3_)	44.50 (25.25, 87.50)	74.00 (41.75, 118.25)	41.50 (23.50, 66.25)	0.010 *
NE (×10^9^/L), M (Q_1_, Q_3_)	20.00 (8.00, 40.00)	20.50 (11.00, 37.00)	19.00 (7.00, 40.75)	0.540
BMB (%), M (Q_1_, Q_3_)	46.00 (28.75, 80.00)	65.00 (30.75, 83.50)	40.25 (28.38, 74.88)	0.241
Spleen Length, M (Q_1_, Q_3_)	97.07 (84.86, 115.19)	102.49 (83.27, 122.87)	96.96 (86.19, 111.58)	0.665
Spleen Width, M (Q_1_, Q_3_)	36.92 (30.91, 43.45)	37.62 (32.40, 43.72)	36.56 (30.83, 43.20)	0.673
CT (HU), M (Q_1_, Q_3_)	131.13 (105.57, 173.06)	177.62 (158.42, 233.25)	113.86 (92.10, 140.50)	<0.001 *

SD: standard deviation, M: median, Q_1_: 1st quartile, Q_3_: 3rd quartile. WBC, white blood cell; Hb, hemoglobin; Plt, platelets; NE, neutrophil; BMBs, bone marrow blasts; CR, complete remission; NCR, non-CR; CT: CT attenuation scores. *: *p* < 0.05.

**Table 2 biomedicines-13-00198-t002:** Univariate and multivariable analysis of complete remission.

Variables	Univariate *				Multivariable *		
	β	OR (95%CI)	*p*		β	OR (95%CI)	*p*
Age	0.00	1.00 (0.97~1.03)	0.866				
Gender	−0.75	0.47 (0.17~1.27)	0.138				
WBC(×10^9^/L)	0.00	1.00 (1.00~1.01)	0.274				
Hb(g/L)	0.03	1.03 (1.01~1.05)	0.009 *		0.03	1.03 (1.00~1.06)	0.094
Plt(×10^9^/L)	0.01	1.01 (1.01~1.02)	0.014 *		0.02	1.02 (1.01~1.03)	0.033 *
NE(×10^9^/L)	0.00	1.00 (0.98~1.03)	0.838				
Bone marrow blasts(%)	0.01	1.01 (0.99~1.03)	0.191				
Spleen Length	0.01	1.01 (0.99~1.03)	0.410				
Spleen Width	0.01	1.01 (0.97~1.05)	0.732				
CT(HU)	0.03	1.03 (1.02~1.05)	<0.001 *	0	0.03	1.04 (1.02~1.05)	<0.001 *

WBC, white blood cell; Hb, hemoglobin; Plt, platelets; NE: neutrophil, BMBs, bone marrow blasts; CT: CT attenuation scores; *: *p* < 0.05.

**Table 3 biomedicines-13-00198-t003:** Diagnostic performance of models in predicting complete remission.

Model	AUC (95%CI)	Accuracy (95%CI)	Sensitivity (95%CI)	Specificity (95%CI)	PPV (95%CI)	NPV (95%CI)	Cut-Off
Nomogram	0.912 (0.842–0.983)	0.865 (0.765–0.933)	0.880 (0.790–0.970)	0.833 (0.684–0.982)	0.917 (0.838–0.995)	0.769 (0.607–0.931)	0.337
Clinical	0.762 (0.639–0.885)	0.743 (0.628–0.838)	0.780 (0.665–0.895)	0.667 (0.478–0.855)	0.830 (0.722–0.937)	0.593 (0.407–0.778)	0.337
CT	0.882 (0.804–0.960)	0.851 (0.750–0.923)	0.880 (0.790–0.970)	0.792 (0.629–0.954)	0.898 (0.813–0.983)	0.760 (0.593–0.927)	155.9

The nomogram was composed of clinical test indicators (Hb and Plt) and CT attenuation scores. Clinical was composed of Hb and Plt in clinical laboratory indicators. The CT model was composed of the measured CT attenuation scores. CI, confidence interval; AUC, area under the curve; PPV, Positive Predictive Value; NPV, Negative Predictive Value; CT: CT attenuation scores.

## Data Availability

Data is unavailable due to privacy or ethical restrictions.

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
