# Peer review of "A Nomogram Built on Clinical Factors and CT Attenuation Scores for Predicting Treatment Response of Acute Myeloid Leukemia Patients"

_biomedicines, 2025, doi:10.3390/biomedicines13010198_

Round 1
Reviewer 1 Report
Comments and Suggestions for Authors
The authors present an interesting diagnostic tool that may be useful in evaluating acute myeloid leukemia cases response to chemotherapy. The study is small with only 74 cases evaluated with a 68% non-response rate. The new diagnostic tool has an acceptable, but not excellent, Sensitivity and Specificity rate, however Specificity has a very wide error rate with a 95% CI width of 0.3.
Overall, the manuscript is confusing as to the timing of CT scans and their utility. Were the scans taken at the start of chemotherapy, and the goal is to determine who will respond to therapy? Or were the scans taken at the conclusion of therapy, and is the goal to use CT scans to replace the conventional bone marrow work up? Also when and how were the hematological parameters measured? The methods do not clearly state if these measures are from traditional phlebotomy or from the bone marrow work up? Also when were the clinical data assessed – at the start of therapy, or at the conclusion? Overall, the timing of these events also may be possible confounders for the prediction model, especially if these clinical features were not evaluated at a consistent timepoint relative to the assessment of CR/non-CR response.
Specifically, Line 18-19: Abstract indicates patients underwent CT “within one week prior to and following the first induction chemotherapy”. So each patient had 2 scans? Or is it that patients had a scan over a range of time, either prior to or sometime after the start of chemotherapy? It is important data to report what was the range of scan dates relative to the start (or end?) of chemotherapy, and if this scan date is a possible confounder in the data prediction modeling.
Additionally, Lines 81-89: Clinical data collected indicates hematological parameters were measured before and after treatment. However the results in Table 1 and Table 2 fail to state if the blood count measures were pre or post-treatment. Clarification is needed regarding the timing of each measure. Data reporting should include what was the range of time relative to the assessment of treatment response.
If the CT scan and blood work were taken at the initiation of therapy, and if the goal is to determine who will respond to treatment, then a further issue is the exclusion of the 15 cases with incomplete treatment course. These cases might be causing bias in sampling if the reason for incomplete treatment was due to disease progression. This might be a cohort worth including as part of a secondary analysis. HOWEVER, if the manuscript is evaluating CT scans and phlebotomy measures taken at the end of therapy in the goal of predicting treatment response, then it is agreed these cases can be excluded.
Other points that should be addressed:
Throughout: The term “Multivariate” is used in error in the paper. Instead the term should be “Multivariable.” See Hidalgo and Goodman, 2013 (PMID: 23153131)
Abstract: Line 17-18: The sample size should be moved further down the Abstract. Revise “Patients with newly diagnosed AML (n=74 total)…” and “Patients were divided into complete remission (CR; n=24) and non-CR (NCR; n=50)) groups…”
Line 24: Please specify what response the model is predicting, complete remission or non remission?
Lines 102-104: Please clarify if radiologists were blinded to treatment outcomes. Were all CT scans rated by both radiologists or only one? If only one radiologist per scan, were the scans randomly assigned to each radiologist? Was the randomization done balanced by CR/NCR status? If both radiologists evaluated each scan, what was the inter-rater reliability and how were divergent conclusions adjudicated?
Lines 113-114; For the construction of multivariable regression models, the methods need to clearly state what was the criteria for inclusion of candidate variables. Also, clarification is needed that modeling is predicting risk of NCR, correct?
Line 129: What are “CT values”? Methods section describes “CT attenuation scores.” Tables 1 and 2 just list “CT”. Please be consistent with terminology.
Line 131: “Spleen thickness” is not indicated on Table 1 or 2. Instead both Tables list “Length” and “Width” which should instead be “Spleen Length” and “Spleen Thickness”. Please be consistent in terminology.
Table 2 needs to clarify if the modeling is predicting odds ratio of treatment response or non-response.
Figure 3 indicates Total Points are corresponding to a “Risk” value. This should probably be revised to read as “Probability of Treatment Response” as risk often implies negative outcomes.
Reviewer 2 Report
Comments and Suggestions for Authors
the presented paper is devoted to study of potential use the clinical blood markers of the patients with acute myeloid leukemia (the levels of hemoglobin and platelets) in combination with computer tomography (CT) attenuation score as complex factor of the therapy prognosis. The authors collected the data from 74 patients, built the nomograms with data on clinical blood markers and CT attenuation score, and demonstrated the increase of the prognostic power. The work is well written and organized. There are a few remarks to be addressed:
1. The specific examples of chemotherapeutics could be added to the paragraph on therapeutic approaches in the Introduction section.
2. The specific therapy received by patients should be described in Materials and Methods, subsection Patients.
3. The whole text should be checked for the proper writing of subscripts.
Reviewer 3 Report
Comments and Suggestions for Authors
In this paper, the authors formulate a nomogram to predict outcome in AML. The paper is interesting and has some element of novelty.
The major issue is represented by the lack of karyotype and genetic features of AML, as well as the type of treatment. Trying to deduce outcomes without these information is futile.
Minor issues:
Acronyms are not used consistently: some are not explained or explained later (e.g. CR / NCR in abstract).
In table 1 CT unity of measurement should be included. Furthermore, it is not very clear what length and width refer to.
Round 2
Reviewer 3 Report
Comments and Suggestions for Authors
The authors responded to the reviewers' suggestions. I understood the goal of the study but I struggle to think of an AML work-up without genomic data in 2019-2023 (enrollment period). Despite the scientific interest, in my opinion, the results could be heavily biased due to the lack of data and thus could be not reliable.
